# Development of Enhanced Vortex-Scale Atmospheric Motion Vectors for Hurricane Applications

**David Stettner [1],\*, Christopher Velden [1], Robert Rabin [2], Steve Wanzong [1], Jaime Daniels [3] and Wayne Bresky [4]**

1   Cooperative Institute for Meteorological Satellite Studies/University of Wisconsin-Madison, 1225 West Dayton Street, Madison, WI 53706, USA
2   NOAA/National Severe Storms Laboratory, 120 David L. Boren Blvd., Norman, OK 73072, USA
3   NOAA/NESDIS Center for Satellite Applications and Research, 5830 University Research Court, College Park, MD 20740, USA
4   I.M. Systems Group, 3206 Tower Oaks Boulevard, Suite 300, Rockville, MD 20852, USA
\*   Correspondence: stettner@ssec.wisc.edu

**Abstract:** Atmospheric motion vectors (AMVs) derived from geostationary meteorological satellites have long stood as an important observational contributor to analyses of global-scale tropospheric wind patterns. This paradigm is evolving as numerical weather prediction (NWP) models and associated data assimilation systems are at the point of trying to better resolve finer scales. Understanding the physical processes that govern convectively-driven weather systems is usually hindered by a lack of observations on the scales necessary to adequately describe these events. Fortunately, satellite sensors and associated scanning strategies have improved and are now able to resolve convective-scale flow fields. Coupled with the increased availability of computing capacity and more sophisticated algorithms to track cloud motions, we are now poised to investigate the development and application of AMVs to convective-scale weather events. Our study explores this frontier using new-generation GOES-R Series imagery with a focus on hurricane applications. A proposed procedure for processing enhanced AMV datasets derived from multispectral geostationary satellite imagery for hurricane-scale analyses is described. We focus on the use of the recently available GOES-16 mesoscale domain sector rapid-scan (1-min) imagery, and emerging methods to optimally extract wind estimates (atmospheric motion vectors (AMVs)) from close-in-time sequences. It is shown that AMV datasets can be generated on spatiotemporal scales not only useful for global applications, but for mesoscale applications such as hurricanes as well.

**Keywords:** atmospheric motion vectors; hurricanes; numerical weather prediction; optical flow; GOES-R; rapid-scans

---

## 1. Introduction

Recent advancements in the constellation of meteorological satellites and their associated sensor technologies have allowed opportunities for creatively improving satellite-derived products used in weather analysis and forecasting. Given the increasing volume and resolution of satellite data now becoming available, it is desirable to seek optimal processing methods to exploit these observations. In parallel, improving data assimilation methods now emerging from the numerical weather prediction (NWP) community are challenging satellite data researchers and providers to advance the quality of their products.

The proper specification and analysis of tropospheric winds is an important prerequisite to accurate numerical model weather forecasts. One type of geostationary satellite data that has been

a critical component in the global observing system is atmospheric motion vectors (AMVs). AMVs are derived from sequential satellite images by tracking coherent cloud and water vapor targets [1], and are an approximation of the local wind at the target height. The retrieval algorithms for deriving AMVs have been evolving since the early 1970s [2]. Most of the major meteorological geostationary satellite data centers around the globe are now producing cloud- and water vapor-tracked winds with automated algorithms using imagery from operational geostationary satellites. The datasets are assimilated routinely in all operational global numerical weather prediction systems and have been shown to produce positive impacts on the accuracy of global model initial conditions and forecasts [3].

Contemporary AMV processing methods are continuously being updated and advanced through the exploitation of new sensor technologies and innovative new approaches. The advances in data assimilation and NWP in recent years have placed an increasing demand on data quality. With remotely sensed observations dominating the initialization of NWP models over regions of the globe that are traditionally conventional data sparse, the importance of providing high-quality AMVs becomes crucial toward realizing superior model predictability.

This is especially pertinent to improving forecasts of high impact weather events such as tropical cyclones (TCs; or better known as hurricanes in our study area encompassing the North Atlantic and eastern North Pacific Ocean basins). Past studies have shown the positive impact that AMVs can have on numerical forecasts of TC tracks using data from previous generations of geostationary satellites [4–6]. More recently, methods to process and improve the quantity and accuracy of AMVs are evolving [7–9]. Higher-spatiotemporal resolution data are being realized through advancing satellite sensors and scanning strategies, increased computing resources for processing the data, and improving derivation methodologies. More frequent dataset availability and improved AMV quality is now possible with rapid image scanning strategies (1–10 min refresh) becoming routine on operational geostationary satellites. Higher spatiotemporal-resolution AMV datasets are now possible over programmable targeted areas (such as TCs) when a "rapid scan" mode is activated by the satellite provider agency.

While these datasets are likely overkill for coarser-resolution global model assimilation systems, regional/mesoscale models can benefit. For example, Ref. [10] documented the impacts of high-resolution AMVs in the operational Australian regional model, and the Japan Meteorological Agency (JMA) found that the assimilation of MTSAT rapid scan AMVs in their mesoscale model with four-dimensional variational data assimilation provided improvements to typhoon forecasts [11]. Several studies used the regional Weather Research and Forecasting (WRF) Model and found positive impact of assimilating AMVs on hurricane forecasts [12–14]. Finally, Ref. [15] indicated that the direct assimilation of high-resolution AMVs has an overall modest positive impact on Hurricane WRF (HWRF) forecasts, but the impact magnitudes are dependent on the (1) availability of rapid scan imagery used to produce the AMVs, (2) AMV derivation approach, (3) level of quality control employed in the assimilation, and (4) vortex initialization procedure (including the degree to which unbalanced states are allowed to enter the model analyses). Therefore, it is important to document the optimal AMV processing strategies for fully exploiting the information content of these data in TC applications, especially given the promise of advanced imagers on new-generation geostationary satellites that are now becoming reality around the globe (e.g., Himawari-8/9 AHI, GOES-R Series ABI, GEO-KOMPSAT-2A/2B AMI, EUMETSAT-MTG FCI).

The motivation for this study is to develop and document methodologies to use the new advanced satellite imaging abilities (such as from the GOES-R Series) to telescope down to the scales of weather systems and derive wind fields that will impact their analyses and forecasts. One of the principal benefits expected from the GOES-R Series is the improvement in the spatiotemporal sampling of images from the Advanced Baseline Imager (ABI). The more rapid image refresh should allow for quantitative improvements in derived products such as AMVs. Another reason we are optimistic that GOES-R Series AMVs can be an important contributor to mesoscale analyses are a result of previous studies using GOES-R proxy datasets [13,14,16–18]. In this study, we build on these pioneering efforts

and take advantage of the advanced GOES-R Series capabilities now available along with new AMV derivation methods. By applying these to the production of higher spatiotemporal AMV datasets with the goal of extracting wind information that benefits TC analyses and forecasts, we can begin to answer questions such as: Can emerging AMV processing methodologies act in unison with very high spatiotemporal resolution geostationary satellite imagery to help accurately depict convective-scale wind flows? Can the resultant AMV fields help diagnostic studies of storm behavior such as path and intensity? And could the enhanced AMV information content benefit rapid-refresh data assimilation for hurricane NWP model forecast improvement?

## 2. Methodologies

AMVs are routinely generated in real-time by satellite data processing agencies around the globe to provide large-scale coverage (near full Earth), with the primary beneficiary being global model assimilation with nominal 6-h cycles. These agencies (geostationary satellites) include the National Oceanic and Atmospheric Administration/National Environmental Satellite Data and Information Service-NOAA/NESDIS (GOES series), European Organization for the Exploitation of Meteorological Satellites-EUMETSAT (Meteosat series), Japan Meteorological Agency (Himawari series), India Meteorological Department (INSAT series), Korea Meteorological Administration (GEO-KOMPSAT series), and China Meteorological Administration (FY-4 series). AMVs over the polar regions are also produced from Low-Earth Orbiting satellites. All of these AMV sources are routinely made available over the Global Telecommunication System (GTS).

Routinely-produced full-disk AMVs by NOAA/NESDIS are processed at hourly intervals from the two operational geostationary satellites (GOES-East and -West). While the operationally-produced AMV datasets are reliable and adequate for global model analyses, the coverage and processing methodologies are not optimized for capturing smaller-scale weather systems. The operational AMV datasets must also pass through the sieve of the National Centers for Environmental Prediction (NCEP) Global Data Assimilation System (GDAS), which employs data thinning and quality-control (QC) strategies commensurate with the purposes of global analyses, and likely does not retain full AMV information on smaller flow scales associated with hurricanes [19]. Regional model/DA systems such as those designed for numerical hurricane forecasting are trending toward nested grids down to cloud-resolving scales. High-impact weather events such as hurricanes may have important mesoscale flow fields that need to be resolved in order to improve these higher-resolution analyses and subsequent model forecasts. Therefore, it is imperative to develop observation strategies that meet these increasing demands.

As noted earlier, one way to locally enhance the coverage, density, and quality of AMVs is by taking advantage of more rapid image scanning, coupled with higher-precision sensors and image navigation, all of which enable improved feature (cloud) tracking. Traditionally, the operational "full-disk" AMV datasets noted above employ image triplets separated by 30 min. More recent satellites allow routine full disk imaging at 10–15 min. Since clouds evolve in time, it is desirable to sample them at the shortest interval possible (assuming solid image navigation and co-registration) to obtain the most coherent AMVs [7]. In our effort to create enhanced AMV datasets over limited/targeted regions, we employ rapid-scan (1 min) imagery taken from the GOES-16 'floating' mesoscale domain sectors that can follow targeted TCs within the satellite's field of view.

The enhanced AMV datasets derived from the processing methodologies described below were generated for selected hurricane cases utilizing fully automated procedures (for a flow chart summarizing these processing procedures, see Figure 1 of [20]) that employ a novel cloud-tracking approach implemented by NOAA/NESDIS for AMV processing in the GOES-R Series era [8]. This approach employs a nested tracking technique with improved vector height assignments, and the quality control (QC) for each derived vector relies on a quality indicator (QI, values 1–100) that is based on a set of coherency and consistency tests. These AMV datasets contain estimates of wind speed and direction as derived from infrared window (IR, 11.2 micron), shortwave infrared (SWIR,

3.9 micron), visible (VIS, 0.64 micron), and water vapor at high cloud top (CTWV, 6.2 micron) imagery (no clear-sky WV vectors were produced for this study given the focus on hurricane central dense overcast enhancement). The VIS (daylight hours) and SWIR (nighttime hours) AMVs are normally only processed to track lower-tropospheric (700–1000 hPa) clouds in the periphery of the vortex circulation void of cirrus clouds; however, one enhancement to the VIS application is described in Section 2.4. The modified processing methodology described below employs GOES-16 rapid scan images at 1-min intervals from a limited-domain, floating mesoscale sector that follows a targeted storm (see example coverage in Figure 1).

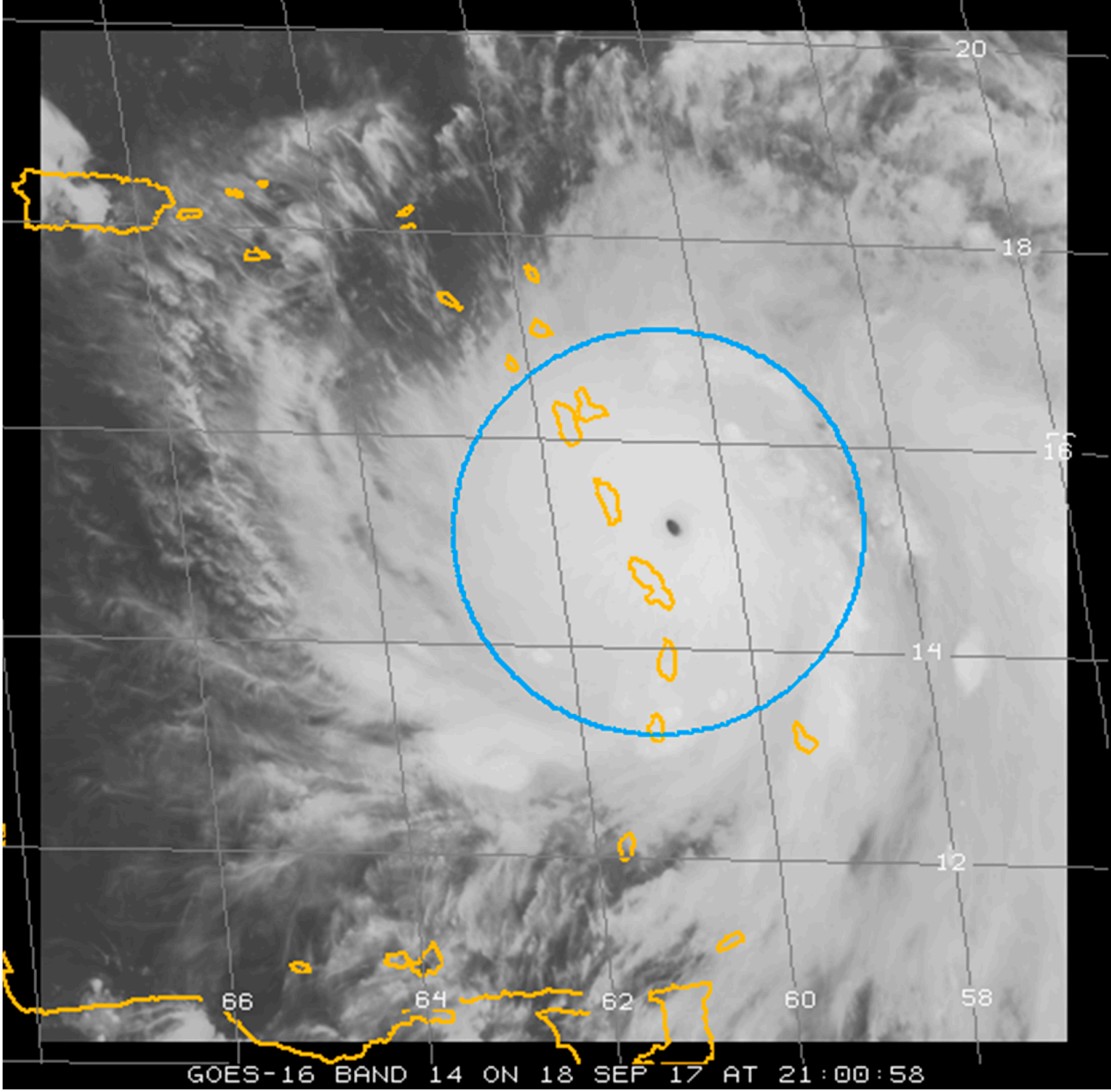

**Figure 1.** Example of the GOES-16 mesoscale domain sector coverage during Hurricane Maria as it approaches the eastern Caribbean Islands. Shown is an IR image on 18 September 2017. The blue circle approximates the storm central dense overcast region as described in the text. Longitudes are plotted as positive West, coastal outlines and islands are plotted in gold.

Two major hurricanes that occurred in the Atlantic Ocean during 2017 (Maria and Irma) were used in the iterative development and analysis of processing strategies to optimize AMV coverage over their near-storm domains, and in particular the central dense overcast (CDO) region (see Figure 1). The resulting processing steps or modifications are described below, often in the context of how they differ from the current operational AMV derivation practices done at NOAA/NESDIS. The AMV processing strategies for enhanced observations around TCs are patterned after previous such studies [5] as a

baseline. Briefly, these strategies include the following: (1) adjusting the target selection, spacing and search box criteria to allow denser AMV coverage to better capture smaller-scale hurricane flow fields; and (2) a relaxation of the QC post-processing steps in the vicinity of a targeted hurricane to allow for the highly convergent/divergent flow fields found with hurricane circulations.

More specifically, the modified processing steps include the following (examples of the impacts of each step will be shown in Section 3).

### 2.1. Increased Target (Cloud Feature) Sampling

The current NOAA/NESDIS operational AMV processing strategy outside the GOES-16 CONUS sector domain looks for potential AMV targets with a spacing of fifteen image pixels. For the GOES-16 IR, SWIR, and WV channels the spatial resolution of each pixel is 2 km at nadir, and for VIS the nadir pixel resolution is 0.5 km. For large scale (full disk) AMV coverage this spacing is sufficient, but in order to capture smaller-scale flow features the spacing between targets is reduced to five pixels. This significantly increases the resulting vector density in cloudy regimes.

### 2.2. Larger Tolerance for Targeting Uniform Cloud Scenes

The hurricane CDO region can have a significant amount of relatively low brightness temperature gradient and coherent (uniform) cloud cover. The operational NOAA/NESDIS algorithm invokes a spatial coherence test as defined in [21] that impacts target selection in very coherent cloud scenes, as well as a brightness temperature gradient needed to define a selectable target. Disabling the coherence test and reducing the required gradient to a minimal level in tandem has the desired effect of improving the identification and tracking of targets in relatively uniform cloud fields such as the hurricane CDO. These changes can be made due to the increased spatiotemporal sampling and precision of the ABI sensors.

### 2.3. More Frequent Image Sampling for Target Tracking

The benefits from more frequent sampling of the triplet of images used in the tracking process have been described above. With the high spatial resolution and signal-to-noise aspects of the GOES-16 imager, the 1 min span between mesoscale domain sector images is more likely to capture and retain highly transient cloud features associated with hurricane flow fields. In our testing of 1, 3 and 5-min image sampling, vector coverage over the hurricane CDO was maximized using the 1 min frequency in all bands processed into AMVs. Qualitatively, the resulting vector fields agreed well with visual image animations of the CDO-top flow fields.

### 2.4. Inclusion of Upper-Level AMVs from Full Resolution VIS Imagery

During daylight hours, the 0.5 km resolution of the GOES-16 VIS imagery allows the detection of cloud structures in the coldest part of the CDO that are not as distinct in the 2 km IR imagery. The relatively small processing domain permits utilizing the VIS images at full resolution, while still meeting real-time processing constraints. The resulting vector height assignments are still provided by the IR window channel derived from the CLAVR-x (Clouds from AVHRR Extended) algorithm which includes the GOES-R Algorithm Working Group Cloud Height Algorithm (ACHA) [22]. Examination of these upper-level VIS AMVs suggests retaining those with IR cloud top temperatures colder than 220 K to avoid mixed-level cloud scenes that can result in erroneous height assignments.

*2.5. Inclusion of Super-High-Resolution AMVs from Experimental Optical Flow Method*

Especially during non-daylight hours, there are still some gaps in upper-level AMV coverage over the CDO region as will be shown in Section 3. Another attempt to enhance this area employs an experimental optical flow (OF) AMV retrieval algorithm that can provide single-pixel spatial resolution. OF methods are showing some promise to derive AMVs [23,24]. Our method employs a "Classical Variational Optical Flow" algorithm obtained courtesy of Thomas Bronx, Department of Computer Science, University of Freiburg and is described in [25]. The algorithm is applied to a pair of GOES-16 IR-window images separated by a 1 min interval. In order to optimize performance, the IR imagery is enhanced to better reveal spatial variations in brightness temperature over the near-uniform CDO. The dynamic range of data contained in the images is then reduced to approximately 205–230 K with a temperature resolution of 0.1 K. The images are remapped to equal latitude longitude grids (plate carrée projection) before applying the algorithm. In the case of GOES-16 data, the grid spacing is 0.02 deg., or approximately 2 km. AMVs are generated at each grid point from the computed displacements. Vector height assignments are provided by the ACHA algorithm.

*2.6. Quality Control Modifications*

Given the dynamic hurricane environment and the relaxing of some AMV processing constraints to achieve enhanced vector coverage, the quality control procedures are particularly important to separate the enhanced signal from the noise in the resulting vector fields. The primary QC method is based on the Quality Indicator (QI) [26]. Results of testing various QI thresholds are not shown below; however, based upon extensive empirical testing and qualitative analysis (actual in situ wind measurements in hurricanes for validation purposes at the CDO-top are virtually non-existent), AMVs are retained with QI values ≥ 50 for all AMVs except for VIS which are ≥90.

## 3. Results

In this section, an example AMV dataset from Hurricane Maria on 18 September 2017 is generated using the NOAA/NESDIS algorithm incorporating the enhancement methodologies described above. We start with the currently operational processing settings, then step through the cumulative impact of the enhancements, illustrating the significance of each modification to the processing. Further interpretations and discussion follows in Section 4. It should be noted that the provided vector plots do not indicate the discrete pressure-level heights associated with each vector since we are emphasizing the general coverage over the high-cloud CDO region. In the forthcoming plots, the vector heights all occur in the 100–250 hPa range.

*3.1. Increased Target (Cloud Feature) Sampling*

Figure 2 highlights the effects of simply decreasing the spacing between attempted targets from 15 pixels to 5 pixels. As might be expected, doing so yields a notable increase in IR and CTWV AMVs as plotted in yellow (QI ≥ 50), especially in the periphery of the hurricane CDO. While some additional AMVs are retrieved in the colder cloud CDO region with this processing modification, the coverage is still quite limited.

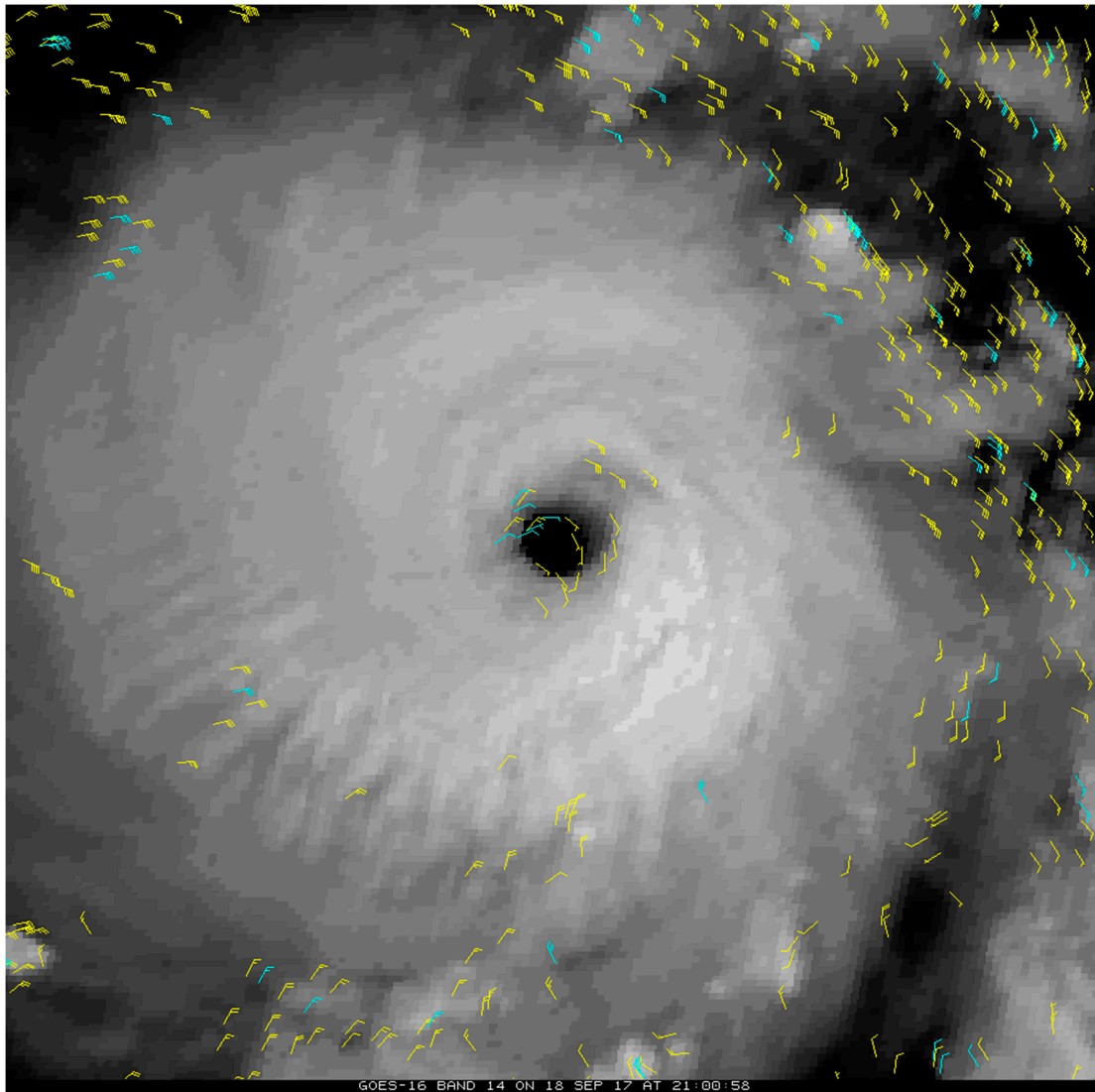

**Figure 2.** Atmospheric motion vectors (AMVs) (IR and CTWV) generated from GOES-16 imagery during Hurricane Maria at 2100 UTC on 18 September 2017. The cyan wind flags were derived using full-disk processing settings such as currently done in NOAA/NESDIS operational AMV production. Decreasing the spacing between targets from 15 to 5 pixels yields the increase in AMVs as shown in yellow. The height assignments for all of the plotted AMVs are in the 100–250 hPa range in this and all following plots.

### 3.2. Larger Tolerance for Targeting Uniform Cloud Scenes

Eliminating the spatial coherence test and easing the required gradient thresholds allows more targets to be tracked in the more uniform CDO region. Figure 3 shows the IR/CTWV AMVs (QI ≥ 50) that are added (yellow vectors) by these processing modifications. There is a notable improvement in AMV coverage in the hurricane CDO region, but big gaps still remain.

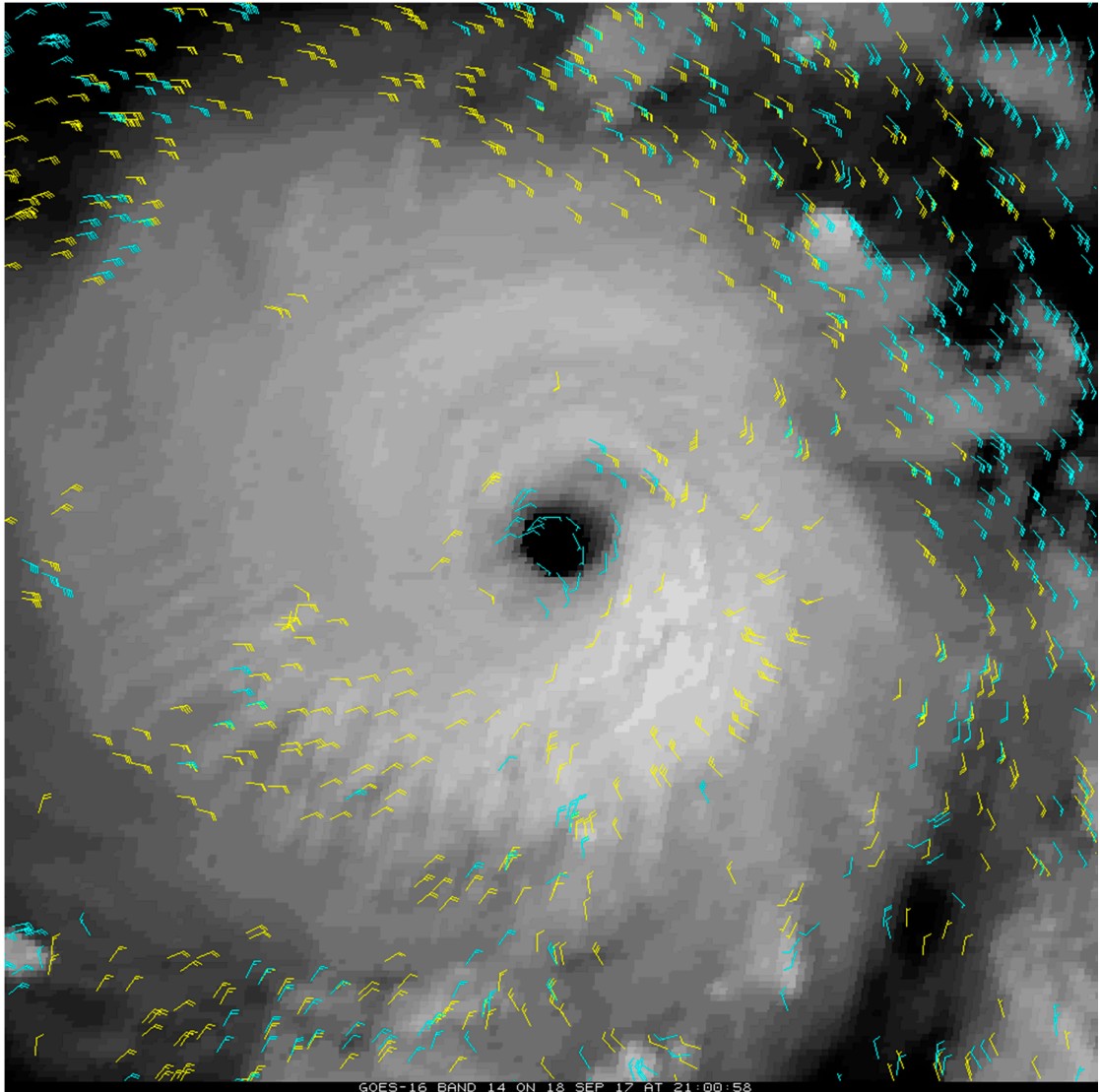

**Figure 3.** Same as Figure 2, except those yellow AMVs are now plotted in cyan. The yellow AMVs here are those added by modifications to the spatial coherence test and gradient thresholds.

### 3.3. More Frequent Image Sampling for Target Tracking

The next step in the enhancement process, described in Section 2.3, is illustrated in Figure 4. The IR/CTWV AMVs (QI ≥ 50) shown in cyan include the enhancements discussed thus far but utilize a 5 min image time step in the tracking triplet (current operational setting for the GOES-16 mesoscale domain sectors). When the time step is reduced to 1 min, the AMVs (QI ≥ 50) plotted in yellow are produced. It is evident that more cloud features have been tracked in the CDO region by employing the 1 min imagery. The hurricane CDO can be a very dynamic cloud regime resulting in rapid morphing of cloud features. Thus, the more frequent 1 min sampling reduces the chance that cloud features will morph, and results in greater AMV coverage. Other differences are noted from Figure 4, and these are discussed in the next section.

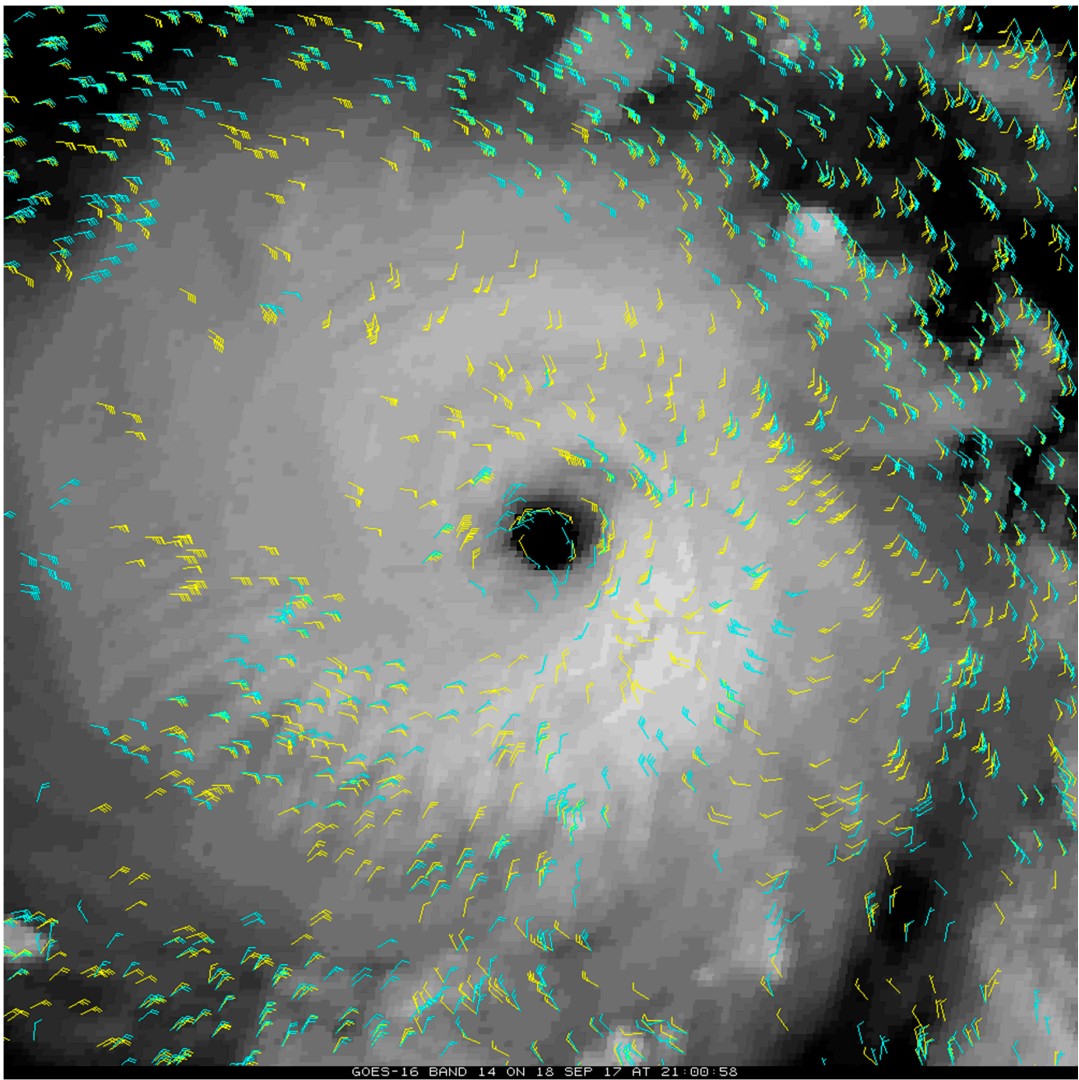

**Figure 4.** Same as Figure 3, except those yellow AMVs are now plotted in cyan. The yellow AMVs here are those added from reducing the image triplet time interval to 1 min sampling.

### 3.4. Inclusion of Upper-Level AMVs from Full Resolution VIS Imagery

While Figure 4 shows that the cumulative effect of the processing modifications leads to a significant enhancement of IR/CTWV vectors in the hurricane CDO region, there are still some areas where further enhancement would be desirable. As noted in Section 2.4, AMVs from visible imagery are normally constrained to tracking low-level cumuliform type clouds. However, the opaque CDO region offers another opportunity for the very high spatial resolution of the VIS images (0.5 km) to detect subtle features that are not as apparent in IR imagery. Figure 5 shows how restricted use (clouds colder than 220 K in the IR) of the full-res VIS can yield additional AMV coverage in the storm core region. Here, the plot represents VIS AMVs (yellow) processed using the cumulative modifications above, but only those with QI ≥ 90. The higher threshold value is necessary to address the increase in noisy vectors due to the inherent subtleties of this extraction methodology. Of course, this application is only available during daylight hours, and the coverage will vary with sun zenith.

Putting it all together, especially in daylight situations, the full set of enhancements can yield a considerable increase in AMV coverage over the hurricane CDO region. Shown in Figure 6 is a comparison of the AMVs available using the operational algorithm settings (tuned for large-scale, full-disk processing) and those possible with all enhancements to the processing strategy as described above.

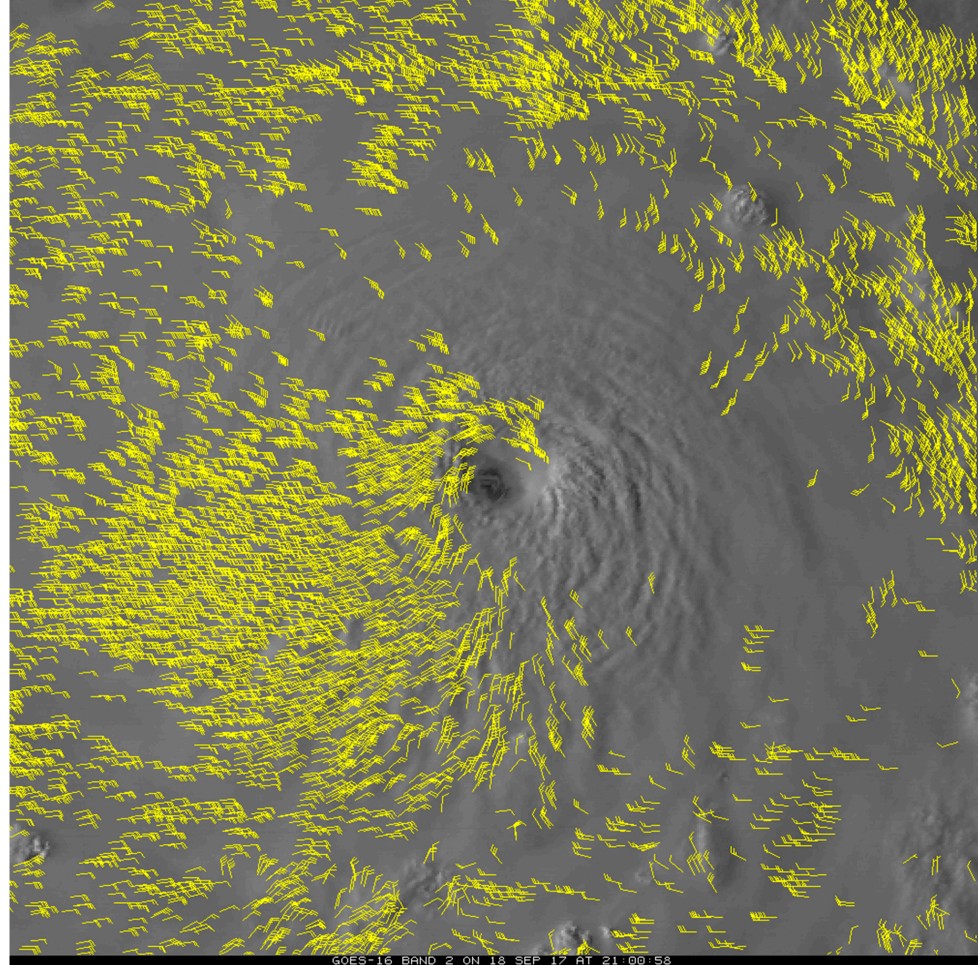

**Figure 5.** VIS AMVs generated for the same date/time as Figure 2 using all of the above modifications for hurricane-scale processing. Plotted AMVs are limited to QI ≥ 90 and cloud top temperatures ≤ 220 K.

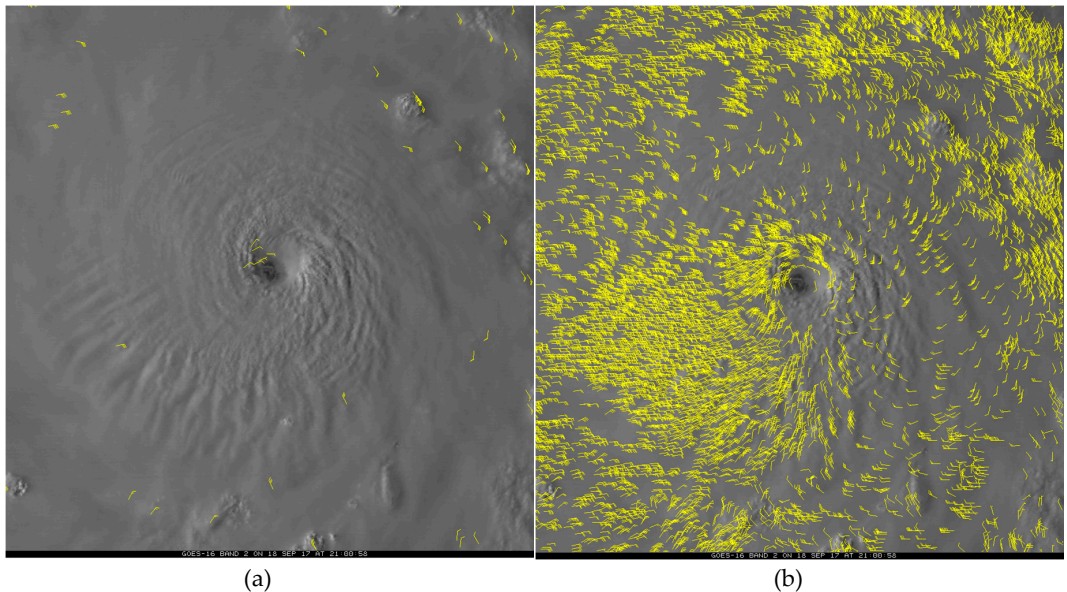

(a)　　　　　　　　　　　　　　　　　(b)

**Figure 6.** GOES-16 VIS imagery of Hurricane Maria (2017), overlain with: (**a**) Upper-level AMVs generated using operational processing settings, and (**b**) AMVs generated using all of the processing enhancements noted above to capture hurricane-scale flow fields.

### 3.5. Inclusion of Novel Super-High-Resolution AMVs in the CDO Core Region from Optical Flow Method

Given the need to have uniform diurnal coverage to the extent possible, an OF algorithm (Section 2.5) operating on IR images was explored as an additional AMV component to the hurricane CDO enhancement. Figure 7 shows the IR AMV coverage that can be achieved by the OF processing method; in practice, the accepted AMVs are limited to cold clouds ≤ 210 K. The nature of the methodology is to produce a vector at each grid point (pixel), thereby producing the continuous field of vectors (the AMVs in Figure 7 have been thinned in the plotting routine in order to distinguish individual flags). While this methodology is still experimental and in need of a quantitative validation, qualitatively the vector field matches other cloud-tracked vectors quite well in most areas and coherently depicts the conceptual model of hurricane flow near the core.

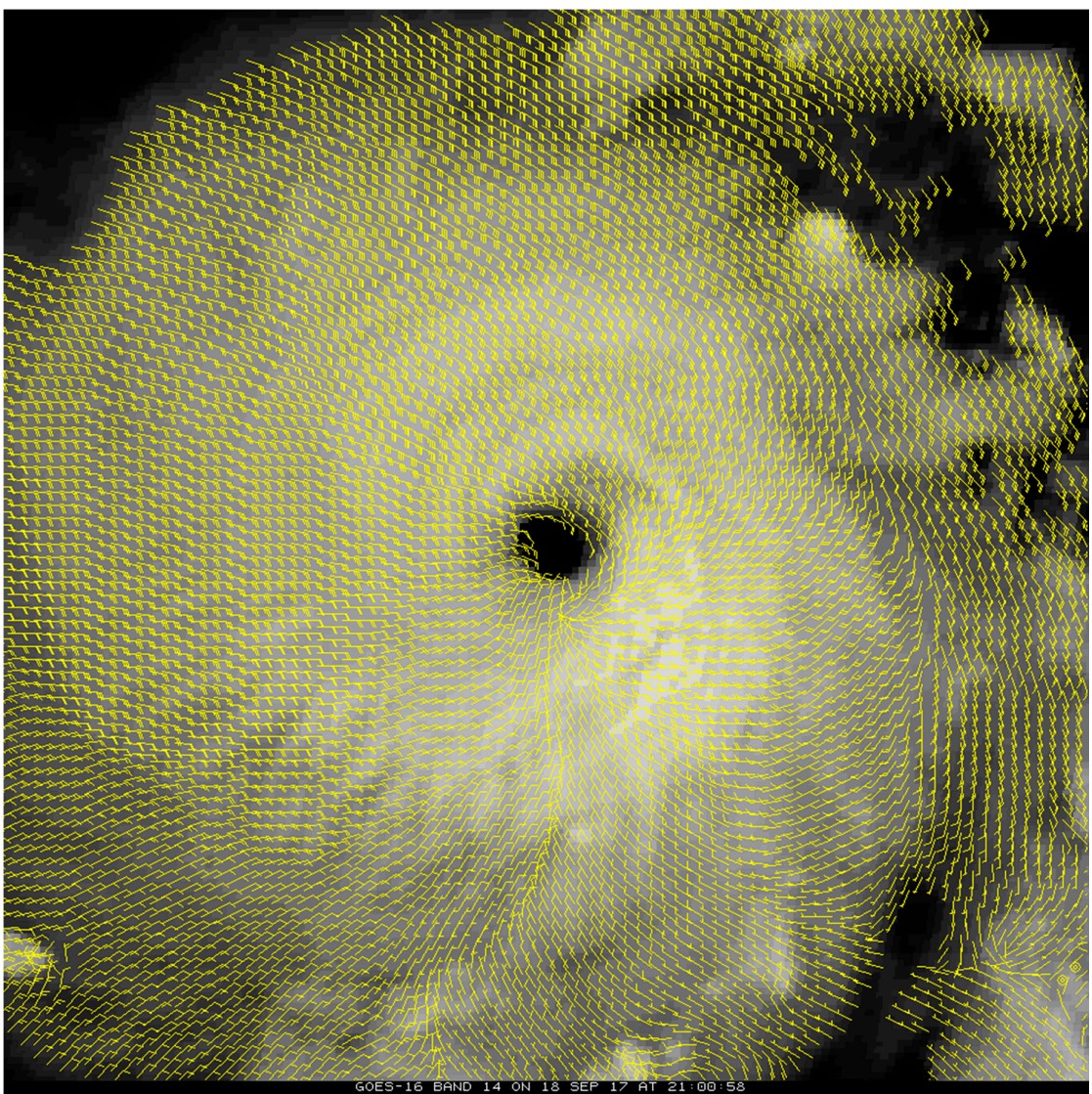

**Figure 7.** Same Hurricane Maria case as Figures 2–6, but plotted are upper-level AMVs generated with an optical flow technique. The wind flags have been thinned in order to make a distinguishable plot.

After the optimization process and assessment was finalized, enhanced AMV datasets during Hurricane Irma were reprocessed at 15-min intervals when GOES-16 mesoscale domain scans were available, which was for most of the storm's lifecycle. An animation of one day (6 September 2017) of these datasets can be viewed in Video S1. Supplemental File referenced here.

## 4. Discussion

The progression of processing steps designed to increase AMV coverage around hurricanes has a cumulative effect as illustrated above. For any individual modification step, gaps in coverage can still exist. However, there is enough independence in the modification procedures so that the various causes of the coverage gaps are addressed in tandem, resulting in fairly uniform CDO coverage when the enhancement steps are combined as shown in Figure 6.

Why is the routinely-produced AMV coverage so poor over the hurricane? As hinted at earlier, the answer lies primarily in the fact that the current operational processing methodologies are not optimized to capture vortex (hurricane) scale flow fields. Traditionally, operational AMV production was principally designed to depict the large/synoptic scale flow for global NWP. Even today, thinning procedures are often utilized in the data assimilation process before the AMVs are incorporated into the global model initial analyses. Therefore, the full-disk processing strategy has focused on quality over quantity (i.e., tighter QC constraints), and adequately capturing the large-scale flow patterns.

So why not apply the enhancement strategies to the routine full-disk processing and then just let users thin the data? First of all, the 1 min rapid scan imagery is not available outside of the floating mesoscale domain sectors. But, perhaps more importantly, it is not practical or efficient for real-time operational processing. Besides hardware limitations and operational processing time constraints, even for research dataset purposes (e.g., reprocessed AMV datasets for re-analyses) the need for the enhanced coverage described in this study is generally limited to dynamic weather systems and/or mesoscale events. Over most of the full-disk imaging domain, this enhanced coverage would be overkill. However, for weather systems such as hurricanes (tropical cyclones, globally), it is important to capture the rapid changes in convectively-forced flow affecting, the development and maintenance of the secondary circulation (inflow/outflow). The GOES-16 mesoscale domain sector coverage is large enough to capture a hurricane (Figure 1), but small enough for routine real-time processing of AMVs datasets with enhanced settings to be generated at 15-min intervals.

With the current state-of-the art hurricane models, 15-min dataset generation is likely adequate. Front-end data assimilation has rapidly progressed, with the HWRF system [27] down to 1 h analysis cycling and able to incrementally adjust for dataset time offsets. As an example of the 15 min sampling, AMV datasets were processed over the lifecycle of Hurricane Irma (2017) during the period that NOAA/NESDIS was targeting the storm with GOES-16 mesoscale domain sectors. An animation of one day worth of these datasets (Video S1) reveals the time-coherent capturing of rapidly evolving flow fields associated with the hurricane and interaction with its near environment.

Video S1 illustrates how the plotted AMV observations could also be used qualitatively by forecasters or researchers to analyze and diagnose storm trends and behavior. More quantitative use might result from integration of the data into objective analyses for hurricane process studies. While AMVs, in general, have been thoroughly validated over the many years of development, opportunities for in situ comparisons especially over hurricane CDO regions are scarce [19]. It can be noted that a few areas of vector disagreements appear in the plots between modification steps in Figures 2–6. Some of these may be due to slightly different targets being tracked, or slightly different height assignments in regions of the hurricane with high directionally-sheared flow. Without independent "truth" observations, it is difficult to say for certain if these differences are explainable.

The enhancement modifications in this study have focused on the hurricane CDO region, which is constrained to the upper-troposphere. However, what about low-level AMVs? In certain TCs, the CDO and cirrus outflow regions are not as dominant as those shown in the Hurricane Maria examples presented in this paper. In these cases, opportunities exist to track clouds associated with the lower-level storm circulation and inflow. While not shown here, some of the processing modifications will also benefit the retrieval and coverage of these AMVs.

Finally, it is shown in Figure 7 that the OF method produces a continuous field of vectors over the CDO region. So why not just use the OF alone in this hurricane application? This could happen at

some future time, but the algorithm is still experimental and the output needs further validation. In the meantime, the cloud-tracked winds methodology is mature and with known error characteristics [28].

## 5. Conclusions

Atmospheric motion vectors derived from multispectral imagery provided by the new generation of geostationary meteorological satellites are important contributors to the tropospheric observing system. With the advanced imagers, we now have the ability to routinely telescope down to the mesoscales with rapid-refresh images and depict flow features that previously were left unresolved. Utilizing a combination of data processing enhancements to existing methodologies and new tools like optical flow feature-tracking, it is possible to generate high-volume AMV fields for applications such as hurricanes.

By applying these procedures to the production of higher spatiotemporal AMV datasets with the goal of extracting wind information that benefits hurricane analyses and forecasts, we can begin to answer the questions posed in the introduction. Emerging AMV processing methodologies can act in unison with very high spatiotemporal resolution geostationary satellite imagery to more accurately depict hurricane vortex-scale wind flows. The resultant AMV fields should help diagnostic studies of hurricane behavior, and we are already seeing the benefits of the enhanced AMV information content in rapid-refresh data assimilation for hurricane NWP improvements [15,29]. While the processing strategies developed in this study will have immediate impacts on research datasets, it was also demonstrated that the procedure could be completed in real-time to be commensurate with operational time constraints.

**Supplementary Materials:** The following are available online at http://www.mdpi.com/2072-4292/11/17/1981/s1, Video S1: 6 September, 2017 Hurricane Irma AMV Animation

**Author Contributions:** Conceptualization, C.V.; methodology, D.S., C.V., R.R., S.W.; software, W.B., J.D., R.R., S.W.; validation, D.S., C.V., R.R.; formal analysis, C.V., D.S.; investigation, D.S., C.V., R.R.; resources, C.V., J.D.; data curation, D.S.; writing—original draft preparation, D.S., C.V.; writing—review and editing, all; visualization, D.S., C.V.; supervision, C.V.; project administration, C.V.; funding acquisition, C.V., J.D.

**Funding:** This research was funded by the NOAA/NESDIS GOES-R Program Office, grant number NA15NES4320001.

**Acknowledgments:** We would like to acknowledge the support of this work from the GOES-R Program Office, the GOES-R Algorithm Working Group, and Steve Goodman of the GOES-R Risk Reduction Program. The UW-SSEC Data Center provided the GOES-16 images for this study.

**Conflicts of Interest:** The authors declare no conflict of interest. The funders had no role in the design of the study; in the collection, analyses, or interpretation of data; in the writing of the manuscript, or in the decision to publish the results.

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
