# Peer review of "Development of Enhanced Vortex-Scale Atmospheric Motion Vectors for Hurricane Applications"

_remotesensing, doi:10.3390/rs11171981_

Round 1

Reviewer 1 Report

This article deals with vortex-scale atmospheric motion vector for hurricane applications. The paper presents the results of the research and clearly shows the advantage of using high temporal resolution satellite data. The results achieved are very nice and are of great importance not only for studying hurricanes.

The structure of the article is clear and both methods and results are clearly presented. In my opinion, the article is almost ready for publication.

I have the following comment. The authors concentrate on the central dense overcast region (line 156) and later they wrote that the plotted vectors occurred in the 100-250 hPa. I am not sure if I understand the text well. Does it mean that vectors in one figure are from different heights? If all vectors within one figure are at the same height, please, say it clearly. If not, I would appreciate a brief discussion whether different heights can cause problems tracking storms or in assimilation or in another application.

Reviewer 2 Report

The manuscript under review is titled “Development of Enhanced Vortex-Scale Atmospheric Motion Vectors for Hurricane Applications”.  It provides an overview of enhanced processing for determination of cloud motion vectors at the tropical cyclone scale.  The paper generally reads well and the narrative flows logically.  Perhaps, it would benefit from a flowchart of the existing algorithm which forms the basis for the new procedures.  I have very few other minor comments which are mostly of a writing/punctuation nature:

Line 88 – I believe that the word “principle” should be “principal”…

Line 145 – I believe that the use of a semi-colon is warranted:  …clouds; however, …and similarly in Lines 170/171, 216/217,

Line 265 – I believe that this sentence can use some help…

Line 281 – comma missing after “situations”, I believe…

Line 326 – comma missing after “But”, I believe…

In the references section, the indentation after the reference number is inconsistent…

Line 151 – the description of the outlines of the coastal regions, etc. (yellow lines) is missing…

Line 174 – the term “relatively low gradient” is missing a reference as to “gradient of what”….

Line 204 – whose “Computer Vision Group”?

Figure 2 and others – the yellow and cyan wind barbs are difficult to discern at times (in my printed version and to my eyes); not sure of a better choice in colors….
